# Public Policies Shaping Mexican Small Farmer Practices and Environmental Conservation: The Impacts of 28 Years of PROCAMPO (1994–2022) in the Yucatán Peninsula

Lesly-Paola Ramírez [1], Birgit Schmook [1,*], Mateo Mier y Terán Giménez Cacho [2,3], Sophie Calmé [1,4] and Crisol Mendez-Medina [5]

[1] Departamento Observación y Estudio de la Tierra, la Atmósfera y el Océano, ECOSUR (El Colegio de la Frontera Sur, Mexico), Av. Centenario km 5.5, Chetumal CP 77014, Quintana Roo, Mexico; lesly.martinez@estudianteposgrado.ecosur.mx (L.-P.R.); scalme@ecosur.mx or sophie.calme@usherbrooke.ca (S.C.)

[2] Consejo Nacional de Humanidades, Ciencias y Tecnologías (CONAHCYT), Av. Insurgentes Sur 1582, Col. Crédito Constructor, Demarcación Territorial Benito Juárez, Ciudad de México CP 03940, Mexico; mmieryteran@ecosur.mx

[3] Departamento Agricultura Sociedad y Ambiente, ECOSUR (El Colegio de la Frontera Sur, Mexico), Periférico Sur y Carretera Panamericana s/n Barrio María Auxiliadora, San Cristóbal de Las Casas CP 29290, Chiapas, Mexico

[4] Département de Biologie, Faculté des Sciences, Université de Sherbrooke, Sherbrooke, QC J1K 2R1, Canada

[5] Instituto para la Gestión del Conocimiento y el Aprendizaje en Ambientes Virtuales, Universidad de Guadalajara, Av. La Paz No. 2453, Guadalajara CP 44130, Jalisco, Mexico; crisol.mendez@udgvirtual.udg.mx

\* Correspondence: bschmook@ecosur.mx; Tel.: +52-983-1377041

**Abstract:** Conditional cash transfer (CCT) programs, generally viewed as policies to modernize and increase agricultural production and commercialization, also have social and environmental impacts. Among the first Mexican CCT programs, PROCAMPO is directed toward traditional agriculture and pays farmers for permanent cultivation, ignoring traditional fallow systems. It was implemented nationally in 1994 to counteract the effects of trade liberalization. Its objectives encompassed modernizing and improving agricultural competitiveness and environmental conservation. Here, we analyze PROCAMPO from the perspective of environmental conservation to understand its effects on agricultural practices and forest cover, specifically in the Yucatán Peninsula, where agriculture sustainability was previously achieved via an alternating cycle of multi-crop system (*milpa*) and forest. We performed an in-depth program analysis, reviewing 51 documents, including scientific literature, technical evaluations, and official records. Research consistently showed direct effects of PROCAMPO on agricultural practices resulting in extensive land use change, including a reduction in crop diversity and the elimination of traditional *milpas* and fallow. PROCAMPO has impacted conservation by causing high rates of deforestation. Our findings show the need to reorient the design and implementation of agricultural policy to increase agroecosystem resilience and ecological service provision to face climate change.

**Keywords:** traditional farming practices; agricultural policies; land use changes; agrobiodiversity

## 1. Introduction

In the 1990s, Mexico experienced a transition from a welfare state to a neoliberal one. During this process, policies encouraged the participation of private companies in the agricultural–livestock sector while a state structure that protected producers and regulated market prices disappeared [1,2]. In January 1994, the Free Trade Agreement (NAFTA) between the United States, Canada, and Mexico took effect, which negatively impacted production in the countryside, mainly due to the import of subsidized grain from the United States [3–5].

As part of a global strategy to counteract the effects of the free market on disadvantaged economies and sectors, conditional cash transfer (CCT) programs have emerged. This type of program differs from traditional social assistance programs in that it provides grants directly to households in exchange for assuming certain responsibilities and the appropriate use of received cash [6,7]. It is under this approach that PROCAMPO, Mexico's direct support program for farmers, was created in late 1993 [8]. The main feature of PROCAMPO as a conditional transfer program is cash payments for farmers who keep their plots in continuous production with a series of eligible crops from a single list applied to the whole country. PROCAMPO has been present in Mexico's 32 states for more than 28 years. It has recorded more than 2 million beneficiaries and 11 million hectares cultivated, according to INAI [9]. Since its promulgation in 1994, its main goals have been the modernization of the countryside and the conservation of the country's natural resources. To achieve these goals, PROCAMPO joined cross-cutting strategies established by the Mexican government involving multiple programs (e.g., PESA[1], PROMAF[2], CNcH[3], and PRODEPLAN[4], among others) and institutions (e.g., SEMARNAT[5] and SHCP[6], among others). These programs and institutions were aligned to each of the six different presidential administrations that kept PROCAMPO active under different names. PROCAMPO has depended on the Secretariat in Charge of Agriculture, which has undergone changes over the program's lifetime[7] [16]. In this article, based on an analysis of the PROCAMPO program, we address the impact that this neoliberal policy for the countryside has had on small farmer practices and the conservation of natural resources in the Yucatán Peninsula.

PROCAMPO has been examined in several studies with regard to its economic aspects, such as the number of beneficiaries reached, hectares cultivated, and yields [17,18]. Unintended effects have also been examined in individual case studies [19,20]. However, there is no regional analysis that captures how PROCAMPO affects local agricultural practices and natural ecosystems that have co-evolved over thousands of years. Here, we address the direct impacts of PROCAMPO on agricultural practices (particularly on smallholder agriculture) and indirect impacts on nature conservation (through deforestation). We focus on the Yucatán Peninsula region, using a meta-analysis of case studies and evaluations of the program from its inception to the present (1994–2022).

The following section describes the study area and the process of data collection and analysis. In addition, we emphasize the importance of the *milpa*[8] as a central system in the research area and its relevance in the analysis of PROCAMPO's effects. Then we focus on the theoretical framework, which addresses conditional cash transfer programs and economic models related to agricultural development in Mexico. We provide detailed explanation of the decades prior to the formulation of specific policies for the countryside in Mexico. This contributes to a better understanding of PROCAMPO, as previous policies laid the ideological foundations for market competitiveness and modern agriculture through technology. These historic events shaped a solid structure that has supported PROCAMPO for 28 years, gaining the acceptance of institutions, governments, and farmers. We show how PROCAMPO has become a complex system, a web of many diverse elements involving a wide network of actors and proposals aimed at addressing the needs of the agricultural sector. We then present and discuss the results obtained after the implementation of PROCAMPO and conclude with a call for a revision of the program.

## 2. Materials and Methods

### 2.1. Study Area and the Persistence of the Milpa

In this study, we investigate the impacts of PROCAMPO, paying special attention to the Yucatán Peninsula. This geographic region, which encompasses the states of Yucatán, Quintana Roo, and Campeche (Figure 1), lies in the extreme southeast of Mexico, extending northward into the Gulf of Mexico and eastward into the Caribbean Sea. Although mostly flat with extensive limestone plains, it also has low hills in the central area. The peninsula is known for its richness in biodiversity, culture, and history [22], as well as its karst topography, marked by the presence of many fractures and faults. Two distinctive features of the

peninsula are the *cenotes* (small water bodies fed by underground water) and *aguadas* (fed by surface water), freshwater sinkholes formed by the dissolution of limestone. These water bodies are of great relevance to the local ecology and have been of historical importance to Mayan civilizations as a source of water supply [23,24]. The climate is warm sub-humid (modified Köppen–Geiger scheme Aw), with two well-defined seasons: the rainy season from June to October and the dry season from November to May. Temperatures are warm throughout the year, with average maximums of around 30–35 °C [25]. The tropical forests range from low, dry forests to evergreen forests, whereas the wetlands range from coastal salty marshes to high mangroves; all offer habitat for a wide variety of plant and animal species, many of them endemic [26].

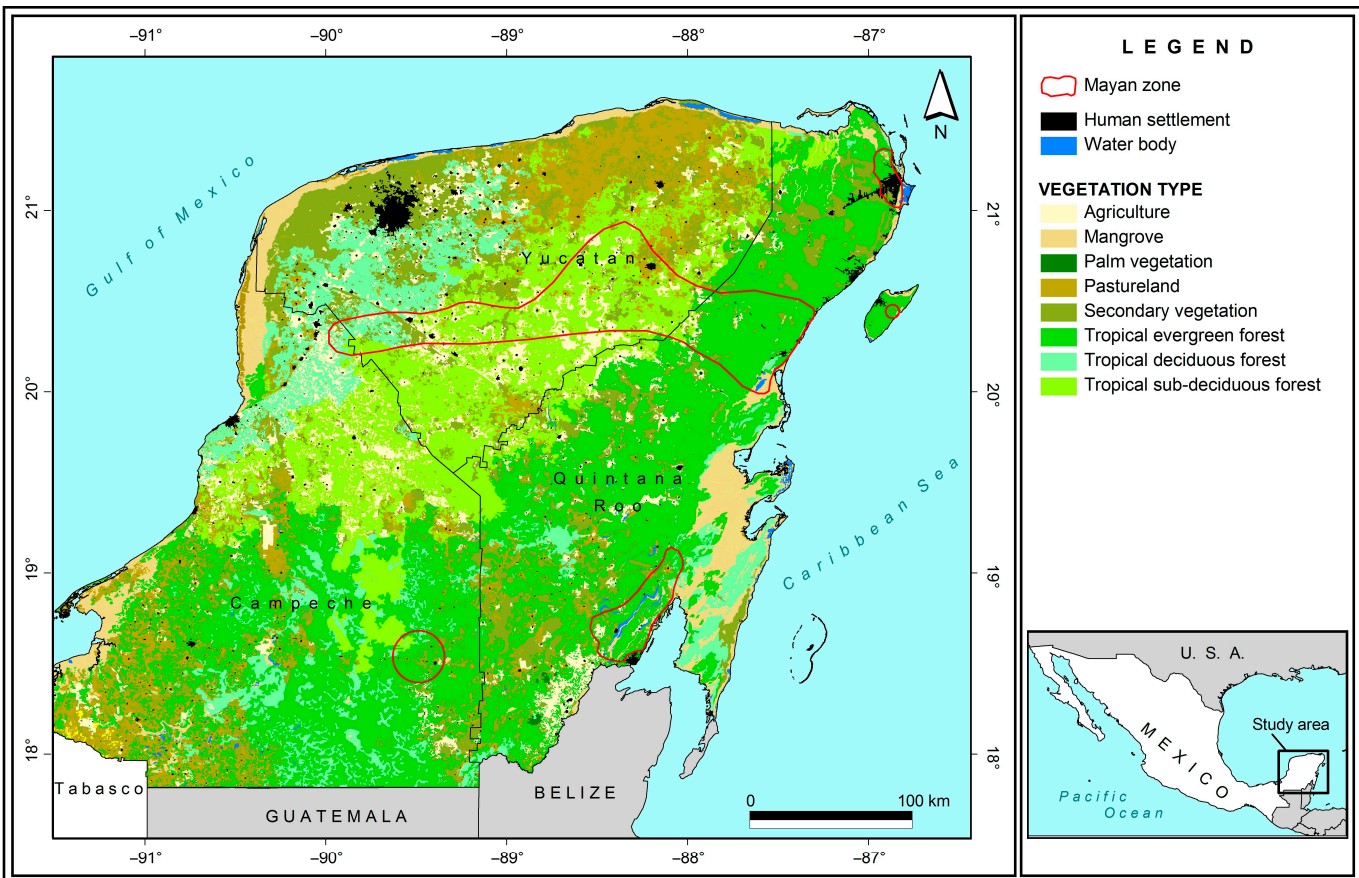

**Figure 1.** The study area, encompassing the Yucatán Peninsula, Mexico.

Traditional agriculture in the Yucatán Peninsula comes from the Mayan civilization, which spread across what is now southern Mexico, Guatemala, Belize, Honduras, and El Salvador. The Maya developed a highly efficient and sustainable agricultural system, the *milpa*, that allowed the Maya to thrive for centuries [27,28]. This system is based on the selection of native seeds adapted to local conditions and agricultural techniques using hand tools, such as slashing and burning [29]. The latter consists of cutting bushes, branches, and thin trees with a *machete* and then finishing removing the larger trees with an axe. Subsequently, the vegetation is gathered and set on fire and then the resulting ash-fertilized soil is planted with crops [30]. Studies in the humid tropics have shown that this system, mostly practiced by indigenous people, affects ecosystems to a lesser extent than modern agriculture [28,31]. The slash-and-burn system consists of a set of millenary practices that include fallows of up to 25 years. Long fallows provide enough time for the vegetation to replenish its productive elements (mostly nutrients). The *milpa* stubble (post-harvest residues) serves as natural fertilizer [28], while some weeds provide soil cover, prevent erosion, and are sometimes used for pest control [29]. Farmers in the Yucatán Peninsula

who cultivate traditionally have small plots in communal land tenure regimes called *ejidos*, dating back from the Mexican Revolution (1910–1917). Their production, consisting mostly of maize (*Zea mays*), beans (*Phaseolus vulgaris*), and squash (*Cucurbita* spp.), is mainly for self-consumption [32]. Most of them are *ejidatarios* (they have property rights within the *ejido*). Traditional maize farmers or *milperos*[9] have been restricted to being subsistence producers because the contemporary market does not return what is invested [35]. At this point, it is important to specify that the transition of traditional farmers' subsistence or semi-subsistence to commercial production is a key element in this study's analysis of PROCAMPO's effects. The program exerted its influence in part by categorizing farmers and assigning differential corresponding payments. In 2003, producers with rainfed land received MXN 1030.00 per ha (USD 95; USD 1 = MXN 11 in 2003), whereas those with irrigated land received MXN 950.00 (USD 82) [36]. In 2009, producers with 5 hectares of rainfed land received what the program called the Alianza quota of MXN 1300.00 (USD 91; USD 1 = MXN 15 in 2009), whereas those with more than 5 hectares of rainfed land received the preferential quota of MXN 1160.00 (USD 80) per ha. New producers or those with irrigated land received the normal quota of MXN 963.00 (USD 67) [37].

The traditional *milpa*, a polyculture system with maize, beans, and squash as its main crops, is the axis of rural everyday life and the Mesoamerican cosmovision and is articulated around issues such as diet, health, religion, and social life [12,32]. According to De Schutter [35], this system should be legally protected because of its productive potential and because of its potential for reducing food poverty. The FAO [38] considers it an "important system of the world's agricultural heritage." It has been demonstrated that the traditional *milpa* system is more sustainable than maize in monoculture due to its production of diverse food crops on the same land, its integral use of local natural resources, the generation of work for farming families, its lower effects on ecosystems, and its contribution to the conservation of agrobiodiversity [39–41]. The *milpa* system is resilient to disturbances (e.g., extreme weather events, diseases, complex socioeconomic and political scenarios) mainly due to its complex agrobiodiversity [42,43]. The *milpa* and its cultivation practices in the Yucatán Peninsula are relevant to the analysis of PROCAMPO, due to the program conditioning payments based on the cropping regimes and crops permitted.

*2.2. Data Collection and Analysis*

The first author conducted a meta-analysis of case studies following Levitt [44] by first identifying a series of primary research studies, converting the primary findings into initial data units for meta-analysis, and developing categories or themes. Data were obtained from a systematic review of articles that analyzed the PROCAMPO program. As a first step, we searched for studies and evaluations at the Red de Revistas Científicas de América Latina y el Caribe, España y Portugal (REDALyC), Web of Science (WoS), Scopus, and Google Scholar from 1994 to 2022 (i.e., from the beginning of the program as PROCAMPO and continuing under different names until the present). The search was undertaken in English and Spanish using the following keywords: "environment," "resilience," "environmental resilience," "cultural resilience," "agroecosystems biological diversity," "landscape diversity," "landscape," "forestry," "agriculture," "family farming," "rural development," "sustainable rural development," "peasant systems," "agricultural policy," "land use change," "deforestation," "tropical deforestation," "Yucatán Peninsula," "rural migration," "sustainability," "public policy," "agri-food," "agri-food system," and "public financing," accompanied by the words "PROCAMPO, Mexico" in parentheses. This yielded 59,618 results in Google Scholar, 448 results in Scopus, and 8 results in WoS. To reduce the number of items found in Google Scholar, we ran a new search with the same terms but only in the titles, which yielded 592 articles.

Subsequently, the number of publications was reduced by manually selecting the most relevant titles. Those with specific information about PROCAMPO in the title and then in the content were selected. Next, publications that only mentioned the word "PROCAMPO" as an example of public policy without providing any information about

the program were discarded. Subsequently, only publications with titles and content describing some consequence or relationship of PROCAMPO in agriculture, practices, conservation, (agro)biological diversity, and vegetation were considered, resulting in a total of 160 publications. These were classified into scientific research that included fieldwork, deskwork, or both; official PROCAMPO documents (decrees, operating rules, manuals, and agreements, among others); and documents on program evaluations and diagnoses. Finally, for the scientific research publications, only those dealing with the Yucatán Peninsula were selected. Our final set included 51 publications, of which 15 were research articles, 29 were official documents, and 7 were technical evaluations.

With the analysis of each document we generated a timeline (see Figure 2). It laid out a temporal axis based on the official documents that contained what was established by PROCAMPO and an axis of parallel events based on program evaluations and scientific research, generating a matrix that sought relationships between PROCAMPO changes and the key themes of this research (public policy, small farmer practices, and biodiversity conservation). In this way, we were able to analyze convergences and divergences in PROCAMPO proposals, implementation, and evaluation. The collection of documents made it possible to condense a matrix with a time frame of 28 years (1994–2022) and six presidential tenures. The analysis of this temporal matrix provided a complete and structured view of the evolution of the program over time and under different federal administrations.

After a first review of the central findings of the documents, we established two variables to analyze the trends found in the meta-analysis: (1) effects on farmers' practices and (2) conservation effects (see Table A1). Subsequently, the first author conducted a thorough review of the documents and coded the information into the following categories: economic dependence on subsidies, land use changes, changes in agricultural techniques, production, environmental degradation, and deforestation. Based on this, we defined indicators by which PROCAMPO's effects on the variables could be explained: land use changes from forest to agriculture, changes from *milpa* to chili and pasture, elimination of fallow, intensification of cultivation, maize monoculture and reduced crop diversity, use of agrochemicals, water pollution, and deforestation (secondary or mature forest). Information from the reviewed documents was grouped in an analysis matrix, and the indicators most frequently mentioned were selected. We then distinguished (1) the direct effects of PROCAMPO on small farmer's practices and (2) the indirect effects of PROCAMPO on conservation. Farming practices were the explanatory variable and conservation was the response variable since the former was assumed to affect the latter. This logical structure was essential to understanding the relationship between variables. Subsequently, the effects of the program on the variables were identified through (a) PROCAMPO official discourse, (b) evaluations conducted on PROCAMPO, and (c) PROCAMPO scientific literature.

Following Levitt [44], we present the results from a qualitative approach, describing the main findings by category and response indicators. We first present the historical analysis of economic models and public policies in Mexico, which allows us to situate PROCAMPO and its changes in different moments of the Mexican context. Subsequently, we present the results of the meta-analysis of scientific literature and evaluations in categories for each variable, effects on small farmer practices, and effects on conservation. We then discuss the contextualized case study findings and trends to generate new insight into PROCAMPO implementation effects.

## 3. Results

### 3.1. Conditional Cash Transfer Programs in Mexico's Development Model for Agriculture

In recent decades, the Mexican countryside has faced an economic, environmental, and social crisis that has affected yield, efficiency, and production in the agricultural sector and has led to high dependence on imports, polarization among producers, and increased poverty and marginalization in rural areas [8,45,46]. Since the 1980s, agricultural policies in Mexico have been increasingly based on paternalism and welfare. Although the paternalistic components of a program often provide training and support for parts of the

target populations (commercial producers) to implement productive strategies, the welfare component relegates less "viable" producers (subsistence farmers) to social assistance programs and support [47]. These policies have generated a marked polarization in the agrarian structure and in the *ejido* system, affecting the ownership and management of agricultural land. This polarization is evident not only in the growing support for big agribusiness but also in the concentration of capital, agricultural subsidies, technology, and resources in general, such as higher-quality land and access to water, among others, for some producer groups [48]. According to Rubio [49], one of the main consequences of this polarization is that subsistence farmers have gone from being exploited to being excluded. This is because farmers unable to compete with the power of subsidized and monopolized markets have been systematically excluded from the country's project. However, despite precarious production conditions and the unequal distribution of government economic subsidies [50], small- and medium-scale producers in Mexico represent 85% of all agrifood producers and generate more than 60% of hired employment, in addition to being the owners and guardians of agrobiodiversity [51].

### 3.2. Economic Models and Public Policies for the Agrarian Sector in Mexico

In this section, we briefly review the different economic models in Mexico's recent past and the ways they have materialized in the different public policy projects for the countryside, up to the moment in which cash transfer programs have acquired a significant role in rural development and agricultural production in the country.

(a)　Primary Exporter

During the period from 1900 to 1929, the expropriation of indigenous lands and increased support for hacienda owners and caciques with more productive lands were prominent in the Yucatán Peninsula [52]. Indigenous *campesinos* (smallholders) were dispossessed of 90% of their land [53]. Under this primary commodity export model, global policies aimed at promoting the export of raw materials to achieve the integration of developing countries into the international economy, resulting in an increasing dependence on less developed countries in relation to the more fully capitalist economies in Europe and northern North America [54,55]. The Mexican Revolution of 1910–1917 also occurred during this period. Although it was initially perceived as a peasant revolution seeking freedom and access to land, once the landowners were dispossessed, the State assumed control and power over land distribution. Chapela and Menéndez [56] argue that this marked the beginning of Mexico's capitalist modernization project. The Ejido Law was enacted in 1920 with the purpose of providing communal lands to indigenous and peasant communities. In 1925, the Banco de México was created, which took control of financial resources. In 1926, the Agricultural Credit Law was enacted and the National Irrigation Commission was established. Subsequently, Mexico's Regional Development Plan was implemented, which aimed to build irrigation systems and expand road and highway networks to connect Mexico's rural areas with major ports and cities. This stage culminated in 1929 during the Portes Gil administration, which stood out for carrying out one of the largest agrarian distributions in post-Revolutionary governments [57].

(b)　Industrialization by Import Substitution

From 1930 to 1950, the emergence of public policies aimed at improving conditions in the rural sector was noteworthy. State efforts were also focused on promoting imports of manufactured and capital goods, as well as generating added value to products [55]. In 1930, the Agricultural Cooperative Bank was established. This phase was characterized by an agrarian land distribution that included higher-quality land, the provision of credits for the countryside, and investments in education, among others [52,56]. Forty percent of that agricultural territory became social or *ejido* property [58].

This period saw a significant change in land distribution in the countryside. In 1930, 80% of cultivated lands were in the hands of *hacendados* (big landowners), but by 1940, this figure was drastically reduced to only 30%. In addition, by 1940, 50% of cultivated lands

belonged to *ejidos*, while the rest belonged to new actors with a growing importance in the agrarian sector: ranchers. This transformation is relevant because, from this moment on, control over production in the countryside remained in the hands of the State, and economic priorities for that production were materialized through different policies for the countryside and various support programs. During this period, an institutional apparatus was developed that benefited the rural sector through actions such as irrigation, improved communications, and other infrastructure works [56]. The social (*ejidal*) sector grew in strength. Small farmer organizations such as the National Campesino Confederation and the Confederation of Mexican Workers emerged, with which alliances were established to implement various public policy programs. Cooperativism was also promoted and strengthened in Mexico during this era [52,58].

In the 1940s, public policy focused on Mexico's industrialization and placed emphasis on the economic development of cities, promoting the expansion of the middle class. As a result, there was a significant and steady migration of peasant labor to Mexico's economically attractive cities and to the United States through the Bracero program. During this decade, two other relevant events occurred in the Mexican countryside: the beginning of a policy aimed at indigenous people with an integrationist goal [59] and the emergence of the Mexican Exporting and Importing Company, which would later become the Compañía Nacional de Subsistencias Populares (CONASUPO), a key actor in agricultural production that would establish the basis for a dynamic of dependency under a paternalistic model of governance [52].

(c)    Monopoly/Oligopoly, Stabilizing Development, and Shared Development

The period between 1950 and 1980 was characterized by a lack of attention to the countryside, despite the existence of some support programs and policies that barely addressed the needs of rural Mexico [52]. The most relevant event during this period was the emergence of CONASUPO, which played a central role in the regulation and distribution of commodity prices, as well as in price control of basic foodstuffs during the following three decades. However, mostly producers cultivating large tracts of land benefitted, since small producers did not meet the requirements for access to the safeguards and distribution of their products by CONASUPO [52,53]. In the 1970s, peasant production began to decline and food prices dropped, resulting in the decline of export production. This period was characterized by reduced government investment in the countryside, a lack of effective policies, the formation of oligopolies and monopolies in agricultural production, and the beginning of a process of disappearance of state enterprises [52].

(d)    Agricultural Liberalization Process

Between 1980 and 2000 the steady transition from a welfare state to a neoliberal state occurred and resulted in the increasing abandonment of the countryside. Government policies encouraged the participation of private companies in the productive sector, leaving it to the market to regulate prices and production as a strategy to promote the modernization of agriculture and economic development of the countryside [52]. In 1994, the implementation of the North American Free Trade Agreement (NAFTA) between the United States, Canada, and Mexico had a significant impact on agricultural production with the import of subsidized grains from the United States [4,5]. As a result, the federal government eliminated price guarantees for most crops, restructured subsidies, reduced subsidized credit for small producers, and decreased investment in agricultural research and extension, favoring private sector provision of agricultural services to competitive producers [1,2]. Small producers were unable to compete, and a few years after the signing of NAFTA, an intense migratory flow of undocumented farm workers to the United States began. Government transfers to the agricultural sector decreased, and agro-industrial production was supported, thereby strengthening large, often foreign, food companies [52]. This period was characterized by the disappearance of CONASUPO; the elimination of public enterprise warehouses; the suppression of guarantee[10] prices for basic crops; the interruption of state support for agricultural supplies and food, including maize tortillas;

and a reduction in credit for farmers [52]. This liberalization process led to a loss of Mexico's food sovereignty and increased migration flows, resulting in an impoverished and depopulated countryside. In response to this agricultural crisis, which started even before NAFTA, the organization known as Apoyos y Servicios a la Comercialización Agropecuaria (ASERCA) was established in 1991. It was created as a decentralized administrative body under the Ministry of Agriculture and Hydraulic Resources with the purpose of supporting the commercialization of agricultural products [61]. In 1993, the Programa de Apoyos Directos al Campo (PROCAMPO) was created, derived from ASERCA, with the objective of increasing the competitiveness of small farmers by providing economic support for cultivation. According to Valencia and his collaborators [52], in practice, these policies promoted clientelism, inequality, and discrimination against small producers.

*3.3. Cash Transfer Programs (1980–Present)*

Since the 1980s, Mexican policies have reflected tensions between free market strategies and measures to mitigate economic hardship, including the signing of trade agreements and agricultural liberalization [62,63]. Neoliberal policies were implemented in response to an economic crisis and pressures by international institutions such as the International Monetary Fund and the World Bank. This included privatization of state-owned enterprises and policies to reduce the size of the government. Conditional cash transfer programs were considered the new silver bullet to help the rural and urban poor become responsible economic subjects and improve their livelihoods.

Cash transfer programs (CCTs) have introduced design features that distinguish them from traditional social assistance programs. First, they provide subsidies directly to households, which changes the accountability relationships between the national government, service providers, and beneficiaries. Conditional cash transfers also allow national governments to establish a direct relationship of dependency with poor families, seeking to foster co-responsibility by requiring families to take responsibility for education, health, and the "proper" use of cash [64].

Eguiarte and Gutiérrez [65] state that conditional transfer programs limit collective action and, as such, have also had an impact on the social organization of small farmers. CCTs have led to the abandonment of old and robust large-scale social protection systems, withdrawing the state from the economy and diminishing public policies to replace them with subsidies in areas such as health, education, and housing. CCTs often consolidate the intensive agricultural model inherited mainly from the Green Revolution, whose negative effects (such as excessive use of chemical inputs) are still present, impoverishing agricultural systems, favoring monocultures and hybrid seeds, and promoting land use change [66,67]. Another effect is that CCTs result in low prices for agricultural products but high input costs, in addition to fostering a loss of biodiversity [68].

Since 1994, PROCAMPO has been one of the main public policy tools in Mexico's agricultural sector and the federal program that has served the largest number of rural residents [16]. PROCAMPO fulfilled one of its central objectives of "transferring resources to rural producers" and formed the largest register of producers in the country. It has contributed to agricultural land management by becoming an economic incentive to obtain formal land tenure [69], in addition to functioning as a credit instrument via advance payments to beneficiaries through the PROCAMPO Capitalization Law in 2001 [70]. In its conceptualization, goals to be achieved through direct support to beneficiaries were established [71], such as the modernization of rural areas and the preservation of the environment by improving producers' incomes, but these goals were not met [69,72,73]. Although the program was originally intended to last 15 years, PROCAMPO has remained a central component of Mexican agricultural policy for 28 years. Over time, it has changed names to adapt to the priority strategies of each six-year period and align with the government mandate, such as PROCAMPO Ecological (2007), PROCAMPO Capitalizes (2007), PROCAMPO to Live Better (2010–2012), PROCAMPO Productive (2013), PROAGRO Productive (2014–2018), and Production for Wellbeing (2018–2022) (Figure 2), updating the same list

of beneficiaries (i.e., maintaining, eliminating, and incorporating producers) through its different rules of operation. PROCAMPO's main objective was "to transfer resources to support the economy of rural producers, who meet the requirements and comply with the conditions established in the Decree and in the regulations issued on its basis" [71]. The decree defined concepts such as "(I) Producers: individuals or legal entities that are in legal exploitation of eligible areas; (II) support: the economic resources transferred by the Federal Government to producers, by virtue of the operation of PROCAMPO; (III) eligible crops: maize, beans, wheat, rice, sorghum, soybeans, cotton, safflower and barley; and (IV) eligible surfaces: to the extension of land that had been sown with any eligible crop, in any of the agricultural cycles, autumn-winter or spring-summer, prior to August 1993." The program support consisted of a single payment per agricultural cycle and hectare sown (or fraction thereof) after the Ministry of Agriculture and Hydraulic Resources had determined that the area was eligible without defining any process for the verification of the resources granted[11]. The causes for which producers did not obtain support were (I) for areas that had been planted in alternation with a perennial crop, sugar cane, or similar or if the eligible crops had been used as a nurse for the establishment of pastures and (II) if the eligible areas exceeded the limits of small property established by the Political Constitution of the United Mexican States and the Agrarian Law. However, PROCAMPO was planned to last for 15 years, so it also contemplated goals such as "increasing the capacity of rural production units through greater participation in the field by the social and private sectors to improve internal and external competitiveness, raise the standard of living and income of rural families and modernize the marketing system through the adoption of advanced technologies and the implementation of production methods based on principles of efficiency and productivity." In addition, it granted definitive rights to the program to producers who registered between 1995 and 1996 [71].

As we will detail in the next section, over time, the way in which cash transfers are conditioned, producers receive technical assistance, and beneficiaries are defined has been modified. Changes in the design and implementation of the program included the introduction of differentiated payments based on producer stratification, as well as the incorporation of new eligible crops and areas with activities and projects that were not eligible. These have resulted in a policy that fails to meet its objectives regarding the viability of agricultural activity. However, it does not resolve structural aspects that limit this viability, which means that the money provided by the program does not solve the underlying problems (e.g., transitions from semi-subsistence to commercial farming) [74]. Instead, these funds become a subsistence income, thus contributing to small farmers' dependence on cash payments.

### 3.4. Effects of PROCAMPO Implementation

PROCAMPO has been one of Mexico's programs with the greatest impact in terms of the number of beneficiaries, permanence through presidential terms, and area cultivated. PROCAMPO's design has set seemingly conflicting objectives, such as the modernization of agricultural production and environmental conservation. In the following section, we analyze the direct effects that the program's implementation has had on agricultural practices in the Yucatán Peninsula, especially on small-scale agriculture. We find that the main effects have been increased reliance on cash payments by beneficiary families, the elimination of the fallow period, and deforestation. We also analyze how changes in peasant practices have affected resource conservation, as they have encouraged the intensification of agricultural techniques, thus favoring soil contamination, erosion, and deforestation.

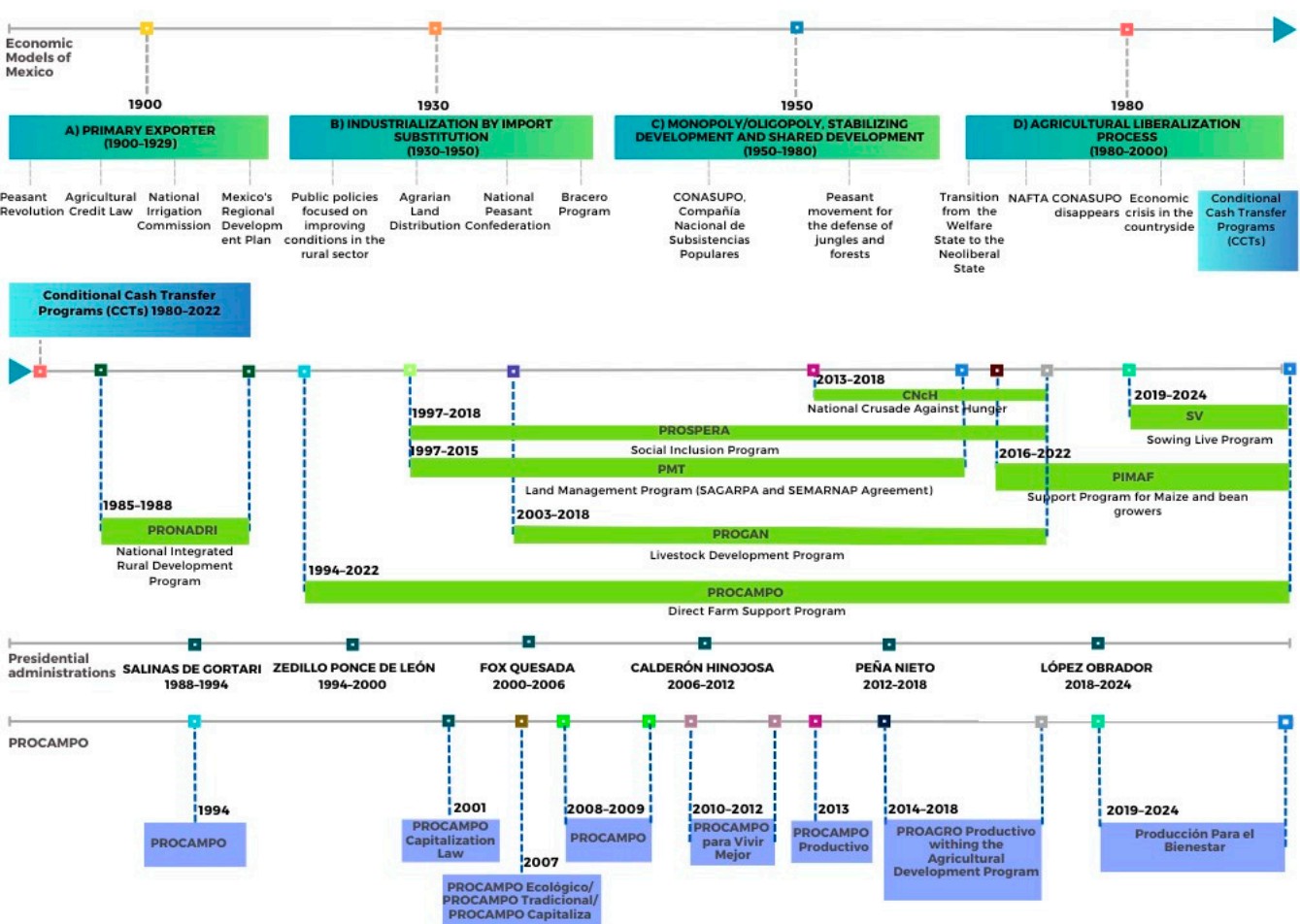

**Figure 2.** The Direct Rural Support Program and its transformations over time in different government processes. Data from [8,75–90].

Effects on Farmers' Practices

(a)　Economic Dependence on Subsidies and Lack of Credit and Technical Assistance

During the first year of PROCAMPO' implementation, farmers received payments if they planted any of the nine eligible crops: maize, beans, wheat, rice, sorghum, soybeans, cotton, safflower, and barley. The decree placed no conditions on the yields to be achieved or how to cultivate the area [71]. In 1994, the payment was MXN 440 (USD 142; USD 1 = MXN 3.10 in 1994) per hectare per agricultural cycle, and beneficiaries could receive this payment as long as they kept their land in production [91]. However, over time, the conditions for receiving the subsidy changed with each new government, including eligible crops, activities considered (livestock, forestry, and cattle), and specific projects (water, conservation, etc.), in addition to the constant redefinition of beneficiary categories, such as those with irrigated, rainfed, ecological, transitional, commercial, or self-consumption projects, among others [72].

At its inception in 1994, PROCAMPO drew its budget from resources previously used by CONASUPO to maintain guaranteed prices. Prior to the implementation of PROCAMPO, CONASUPO guaranteed producers a marketing channel by purchasing a large portion of the national production of certain crops at guaranteed prices. CONASUPO used to purchase more than 40% of the national production of corn, 39% of beans, and 21% of wheat. However, when CONASUPO disappeared, PROCAMPO did not implement specific measures to modernize the marketing system and focused on providing direct cash transfers to beneficiaries. This situation fostered dependency on program payments [72].

During the early years of PROCAMPO, extension services for farmers were not included along with cash payments [70]. In 2007, agricultural extension was incorporated into the rules of operation; beneficiaries would receive services and, in return, were required to submit quarterly production reports [92]. Subsequently, extension services were again explicitly included in 2015 as a "right of PROCAMPO beneficiaries called PROAGRO Productivo in this season (2014–2018)" [93] and have endured to date through the Production for Wellbeing program [94].

Several studies conducted in the Yucatán Peninsula on the implementation of PROCAMPO, such as Klepeis and Vance [95] and Radel, Schmook, and Chowdhury [96] found that the absence of technical assistance from the program and the lack of access to credit increased the costs of the farmers' transition to a semi-subsistence economy in order to access markets, which, coupled with the production constraints faced by farmers, such as water scarcity and rainfall variability, led to a heavy dependence on cash transfers. Schmook and Vance [63] found that, although PROCAMPO technical assistance and funding were primarily aimed at increasing food production, the predominantly *milpa*-based farming population was unable to cope with the development of government-sponsored grazing livestock projects in the area (see next section). This largely occurred because cash payments did not provide sufficient capital for making improvements to the land that would sustain competitive production for small farmers.[12]

PROCAMPO failed to increase staple crop production or transform subsistence (and semi-subsistence) producers into commercial farmers. As a result, many found themselves in an economically unsustainable transition and increasingly relied on cash transfers to stay afloat. In 2006, PROCAMPO accounted for 100% of the income of 52.8% of the beneficiaries surveyed [69]. Cash payments from the program replaced income derived from subsistence farming, and PROCAMPO money went mainly to cover basic food needs and to invest in material goods.

(b)　Land Use Change: From the Agrobiodiverse *Milpa* to Monocultures and Pastures

As mentioned above, PROCAMPO established, as part of its 1994 operating rules, the nine crops eligible for payment. These crops were kept in the program until 2000 [98], and subsequently, different crops were added, reaching a total of 27 [94]. Starting in 2009, PROCAMPO implemented a segmentation in payments based on the specific area planted. For example, (I) Alianza quota (up to 5 ha of rainfed land) with a payment of MXN 1300.00 (USD 96; USD 1 = MXN 13.50 in 2009) per ha, (II) preferential quota (more than 5 ha of rainfed land) with a payment of MXN 1160.00, and (III) normal quota for the rest, both for new and irrigated beneficiaries, with a payment of MXN 963.00 [37]. This ongoing recategorization of quotas and farmers led to as many as six different segments in the program's operational rules in 2017 and 2022 [94,99]. In 2019, PROAGRO Productivo (see previous section) changed its name to Producción para el Bienestar [100]. In this new phase, small- and medium-sized producers with up to 20 hectares, dedicated to grain cultivation, such as maize, beans, wheat, and rice, among others, were included (in previous years, small-scale producers were only allowed to have up to 5 hectares). The operating rules introduced the concept of the *milpa* system, which is defined as "a traditional rainfed agricultural production that combines different crops for family food" [100]. In subsequent years, the program included new eligible crops, such as amaranth (*Amaranthus*), chia (*Salvia hispanica*), coffee (*Coffea*), sugar cane (*Saccharum officinarum*), flour wheat (*Triticum aestivum*), and cocoa (*Theobroma cacao*), and even introduced foraging area for honeybees to the list [94,101].

As originally promulgated, PROCAMPO established that program payments could facilitate the conversion of land to more profitable activities [71]. In 2001, the PROCAMPO Capitalization Law established that productive projects should be directly related to primary production, agro-industrialization, and input supply, or linked to other agricultural, forestry, and fishing activities [70], which encouraged changes in land use that could include deforestation.

Mardero et al. [46] found that, from 1980 to 2015, irrigated (monocrop) maize area increased steadily in the Yucatán Peninsula. Ellis and Porter-Bolland [102] noted that PROCAMPO support was leading to changes in agricultural practices in some localities in Quintana Roo, where crops such as sorghum, sugarcane, and corn grown in monoculture were being introduced, replacing the Mayan *milpa*.

From 2001 to 2006, PROCAMPO's specific objective was the nationwide promotion of grazing areas. Pastures require less investment, which was important in the face of the decapitalization of productive units, the lack of infrastructure and human capital, missing financing schemes, and soil exhaustion and deterioration, the main factors in the problem of low productivity of food crops that the program sought to solve [73]. However, the effects of these program intentions had already been felt locally in the Yucatán Peninsula. In this regard, Klepeis and Vance [95] found that, in the Yucatán Peninsula, the program was associated with a decrease in forest area and an increase in pasture and cash crops. The study showed that the implementation of PROCAMPO in 1996–1997 contributed significantly to the increase in chili cultivation and the creation of pasture areas while promoting deforestation through the clearing of mature forests. Likewise, in a program evaluation, SAGARPA and FAO [72] found that, from 1995 to 2009, perennial grasses ranked fourth in the designation of areas supported by the program after corn, sorghum, and beans. ARAPAU and Associates [69] found that, in the year 2009, perennial grasses stood out as the most beneficial culture and occupied the largest area within the PROCAMPO program. Of the total area allocated to different types of crops, including those that historically were not eligible for the program, such as oats, alfalfa, sugarcane, and agave, grasses accounted for 8%. This translated into a total of 144,000 producers receiving support and covering an area of 1.08 million hectares. According to ARAPAU and Associates [69], the increase in cultivated pasture areas was explained by the campaigns carried out by SAGARPA to remove (abandon) plots that were planted with crops that cause heavy erosion, such as beans in Zacatecas and sorghum in Tamaulipas. The authors also noted that, "The increase in pasture areas and, therefore, the increase in PROCAMPO support to beneficiaries allowed them to grow perennial pasture exclusively without incurring additional costs associated with other crops. It also allowed them to comply with the program's requirement to keep their land planted year after year in order to receive PROCAMPO support" [69] (p. 108).

Since its beginning, PROCAMPO has been associated with a decrease in forest area and an increase in cash crops and pastures in the Yucatán Peninsula [95]. For example, Turner II and colleagues [103] found that PROCAMPO subsidies encouraged some farmers to deforest and establish pastures after growing corn, although approximately 50% of these pasture areas was not being used, as even farmers without livestock planted pastures just to receive PROCAMPO money. Likewise, Ellis and Porter-Bolland [102] reported that, from 2000 to 2005, PROCAMPO was one of the main factors in the process of agricultural expansion in the Yucatán Peninsula. They reported that productive reconversions in Campeche occurred through the elimination of *milpas* to convert them into pasture for cattle. Ellis and collaborators [104] also identified PROCAMPO as one of the most important factors in land use changes in the Yucatán Peninsula from 2001 to 2013, alongside PROGAN. For instance, in Campeche, from 2001 to 2013, sisal plantations and cornfields were abandoned and converted into pastures through PROCAMPO [104]. Collectively, the mentioned studies provide a clear perspective on the impact of PROCAMPO as a key factor in land use changes in the region, suggesting a profound relationship between politics and landscape transformation in the Yucatán Peninsula.

(c)     Elimination of Fallow Land and Intensification of Cultivation

In its decree [71], PROCAMPO proposed the "adoption of modern agricultural technologies and modes of production considered efficient and productive." It also established that "eligible areas should be maintained in continuous production in order to continue receiving program payments in subsequent cycles." In 2001, the PROCAMPO Capitalization Law, which anticipated program payments until 2009 [70], was able to encourage the continuous cultivation of land reported in the program's registry. The fact that PRO-

CAMPO gave economic support based on the area planted (and not as a fixed amount to the producer) encouraged small producers to move to more extensive forms of production (such as pastures or less agrobiodiverse *milpas*). Farmers responded to what PROCAMPO qualified as more profitable activities, as measured in more area planted per year [69]. Further, several studies demonstrated that the implementation of PROCAMPO in the Yucatán Peninsula in the late 1990s changed the traditional fallow cycles that characterized the *milpa* because of the requirement for the land to be in continuous production [95,105]. Some of the decrease in fallow periods among families who had received PROCAMPO money since its inception was explained by diverse interpretations and forms of implementation at the local level [106]. For instance, farmers interviewed by these authors explained that program payments were only granted for "fallow cleared *milpas* and secondary forests," although this was not officially stated in any PROCAMPO document. However, the very financial incentive provided by PROCAMPO also explained the decline of fallow and the loss of forest. Fallow with legumes, for example, is characterized by the non-use of external inputs, making traditional systems such as *milpa* more sustainable [107]. Even short fallows of two to four years using legumes contribute to the recovery of soil fertility by improving the contents of organic matter, NO3, K, and Mg [108]. Schmook and Vance [63] showed that, in the Yucatán, the subsidies received from PROCAMPO between 1997 and 2003 corresponded to the loss of 0.196 hectares of forest per MXN 100 of subsidy received. The program influenced the elimination of forest fallow by conditioning beneficiaries to keep their plots in productive use, removing their land from the forest fallow cycle that allows for the maintenance of soil fertility. This in turn pushed beneficiaries to convert more forest land to compensate [63]. The PROCAMPO payment system was incompatible with the forest–seed cycle, the predominant practice in the region by most inhabitants, which traditionally included a fallow cycle with mature forests of more than 20 years [95].

*3.5. Conservation Effects*

In its initial decree, PROCAMPO included environmental conservation and stated that it was necessary to contribute to forest conservation and reduce soil erosion and water pollution [71]. In 1996, PROCAMPO operating rules stated that the acreage of already registered beneficiaries would be eligible if they had an ecological project authorized by SE-MARNAP, complying with agricultural, livestock, forestry, and water use regulations [109]; however, these regulations are not specified in PROCAMPO's new operating rules and are open to interpretation (2009–2012 and 2019–2022).

Despite these efforts to include this environmental conservation component discursively, the program had negative environmental impacts, including deforestation, during its implementation [110,111]. For example, Ellis and Porter-Bolland [102] showed that the reduction or elimination of fallow periods affected forest cover and forest recovery. These impacts have shown that PROCAMPO subsidies have, to a large extent, had harmful indirect effects and therefore have not contributed to the protection of natural resources in the context of environmental policy [112]. By incentivizing beneficiaries to keep registered landholdings in permanent cultivation [71], PROCAMPO forced producers to plant in any way possible, on marginal rainfed land and using chemical inputs (sometimes supplied by the program itself) to maintain production [73] and not lose their registration in the program.

(a)    Use of Agrochemicals for Agricultural Intensification

The 2001 PROCAMPO capitalization law stated that an advance payment would be granted to productive projects that contributed to more sustainable water use, conservation, and improvement of natural resources and environmental services [70]. In 2007, the government promised to stop the advance of the agricultural frontier and changed the name of the program to PROCAMPO Ecológico, "which from that moment on would be focused on the sustainable management of native ecosystems that provide environmental services, such as soil protection" [113]. In contrast to this goal, in 2001–2006, PROCAMPO proposed an objective "to promote agricultural production through the use of modern

technologies for cultivation, the exploitation of water resources and the identification of markets." By 2016, the program's operating rules supported producers with a wide range of agrochemicals to improve productivity through its sister program, PROAGRO Productivo [73].

These seemingly contradictory goals led to well-documented environmental degradation. In an evaluation by SAGARPA and FAO [72] that addresses the problems of PROCAMPO from 1990 to 2009, soil degradation and overexploitation of water resources were identified as the main effects of the program. Likewise, a later evaluation made by FAO and SAGARPA in 2015 [114] reported overexploitation of aquifers, low efficiency in their use, and excessive contamination. The evaluation found that the availability of quality seeds was limited and that there was a high external dependence on fertilizers.

Radel, Schmook, and Chowdhury [96] noted that, in the Yucatán Peninsula from 1997 to 2003, the transition from subsistence corn cultivation to more intensive corn production over a larger area had environmental implications due to the indiscriminate use of agrochemicals and their runoff and the resistance of multiple weeds to herbicides, with likely consequences for human health and biodiversity conservation. Winters and Davis [115] also observed the indiscriminate use of agrochemicals and their harmful effects associated with PROCAMPO at the national level in 2008. Farmers in Calakmul mentioned that the use of agrochemicals is partially associated with a prolonged cultivation period and the postponement of fallows. Many farmers considered the use of agrochemicals (fertilizers, herbicides, and pesticides) essential to achieving this prolonged period and thereby meeting their production objectives. The use of agrochemicals was introduced in the region between 1980 and 1990, along with the cultivation of chili, but farmers eventually extended their use to maize [116]. PROCAMPO can be directly linked to the increased use of agrochemicals to maintain crop production as a strategy to compensate for the reduced fallow period and to continue receiving program payment for cultivated plots.

(b)   Deforestation

Deforestation was one of the unintended negative consequences of the implementation of PROCAMPO. Ellis, Gómez, and Romero-Montero [117] found that PROCAMPO-driven agriculture was the prime cause of forest loss in the Yucatán Peninsula after 1994. Specifically, Klepeis and Vance [95] found that staple crop production accounted for approximately 70% of the increase in deforestation that took place after 1994 in the Yucatán Peninsula and concluded that PROCAMPO was responsible for 38% (234 ha) of the deforestation in their sample of 609 ha of forest converted to agricultural use from 1994 to 1997. Also, Abizaid and Coomes [105] concluded that, in 1998, PROCAMPO induced logging because the allocation of its per-hectare transfers resulted in beneficiaries accumulating land to access more money from the program. Furthermore, farmers began to cultivate fallow plots because PROCAMPO only supported fallows after *milpa* cultivation during the first year of registration, and thereafter they had to be kept under continuous production [118]. This not only incentivized the elimination of fallows but also had negative effects on the *milpa* system and encouraged program beneficiaries to cut down secondary forests up to 15 years old [119] to access cash transfers.

For their part, Krylov and collaborators [120] found that, in the Yucatán Peninsula from 2000 to 2012, the permanent loss of vegetation cover was mainly due to the planting of grasses, corn, henequen, and sugarcane, which were incentivized by PROCAMPO. Ellis, Romero Montero, and Hernández Gómez [104] also identified PROCAMPO as the underlying cause of deforestation and land use change in the Yucatán Peninsula. Previous studies have shown that these changes in land use have an impact on the composition of microbial communities, which in turn modifies soil fertility, a critical factor for agricultural productivity and yield [121–123]. Mechanized commercial agriculture also contributed significantly to increased deforestation in both northern and central Campeche and southern Quintana Roo [106]. Finally, the promotion of oil palm (*Elaeis guineensis*) by the Alliance for the Countryside Program and PROCAMPO encouraged agricultural expansion and contributed significantly to deforestation in eastern Tabasco [111]. These authors observed

that PROCAMPO beneficiaries had a smaller proportion of forested area on their farms compared to farmers receiving other types of support.

In 2015, PROCAMPO was embedded in the Program for the Promotion of Agriculture, and its rules specify that PROCAMPO transfers were intended for areas with agricultural production and that they should reflect a sustainable use of natural resources. In 2016, the Promotion Program withdrew this specification to focus only on the productivity of the plots it called "rural economic units" [73]. Likewise, in 2019, the operating rules of the Production for Wellbeing program (formerly PROCAMPO) established that the main objective of the program would be to provide liquidity to small- and medium-sized producers for production with a "sustainable" approach [100]. However, as of 2022, there has been no mention of any action that incentivizes conservation in the program's current operating rules.

(c)     Environmental Degradation

The implementation of PROCAMPO by eliminating guaranteed prices generated greater competition between imports and domestic production. This, in turn, increased pressure on farmers so that, far from slowing degradation, PROCAMPO led to "greater degradation of natural resources because of their use in agricultural activities and lower productivity" [71]. This evaluation identified several influencing factors, such as the lack of modernization and appropriate technology, as well as the overexploitation of soils, which caused the loss of vegetation cover and nutrients, negatively affecting the quality (fertility) of soils. By 2007, the Agricultural, Livestock and Forestry Census [124] found that approximately 14% of production units that year faced significant problems related to soil quality for crop development in Mexico. In Colima, for example, the program led to soil erosion, as it caused farmers to work the same plot continuously to continue receiving support, which wore out the soil and reduced its fertility [110].

## 4. Discussion

Our main findings are that PROCAMPO has, directly and indirectly, affected the agricultural practices of its beneficiaries as well as land conservation in the vicinity of the Yucatán Peninsula. The modification or elimination of some traditional agricultural practices, such as the shortening of fallow periods and the transformation of *milpa*, traditionally a polyculture system, into a maize monoculture, is one of the most visible changes [72,95,96]. The establishment of new agricultural lands subsidized under the PROCAMPO program has led to increased deforestation that threatens the conservation of forests and the services they provide to humans, including agriculture. The condition of agroecosystems and surrounding ecosystems has a direct impact on the social and economic situation of small farmers [125] and is critical for sustainable food production and regional food self-sufficiency [126]. Fallow periods, especially long fallows, are not only ecologically important to the functioning of slash-and-burn agriculture because they give the soil the time it needs to recover and prepare for the next growing season but also necessary for vegetation to recover and maintain ecosystem services [28,127,128]. PROCAMPO's impact on land use, intensification of farming practices, and soil contamination and erosion demonstrates that the impact extends beyond cropland and makes urgent the need for significant structural changes in the program's design and implementation.

PROCAMPO was implemented as a federal program without considering the agricultural variability and regional farming characteristics of the country, nor the economic, social, and technical differences between subsistence small farmers and commercial farmers. This "one size fits all" approach resulted in payments to small farmers and commercial farmers based on acreage and for crops specified in the program, based on the simple assumption that beneficiaries would thereby increase their production and simultaneously contribute to the conservation of natural resources. The resulting homogenization of agricultural practices within the national territory was particularly evident in Yucatán, where traditional cultivation techniques were transformed within a short period of time (1994–1997). For example, the traditional *milpa*, a polyculture system consisting at least of corn, beans, and

squash, has been reduced to corn monocultures or even converted to pasture. Not only has this simplification resulted in less diversity of food crops but the loss of agrobiodiversity will also affect the resilience of the cropping system [39–41,43]. Functional diversity—in this case, agrobiodiversity—increases the resilience of systems and provides the ecosystem services on which agricultural activities depend [129]. The resilience of agroecosystems is directly related to the economic and social resilience of small farmers, who, by engaging in sustainable agriculture to ensure food security for their families, also contribute to regional food self-sufficiency [126]. Because of the linkage between agroecosystem resilience and small farmers' economic and social resilience, factors outside of these systems, such as CTC payment and the resulting obligations, can alter these interactions and lead to severe negative direct and indirect effects [130,131], which include the loss of agrobiodiversity and a reduction in systemic resilience in the Yucatán Peninsula.

PROCAMPO has provided limited technical assistance to beneficiaries, and the impact on the small farmer sector in terms of the modernization of agricultural production was achieved only by a small number of small farmers subsidized by the program. The money paid to beneficiaries by the program has translated into an opportunity to access inputs, such as agrochemicals, which promotes dependent production systems. These inputs, which have been conditionally inserted into the practices of small farmers, are insufficient to promote a significant increase in agricultural production, and they exclude native seeds and the agrobiodiversity of traditional agriculture. Most small farmers continue to produce for subsistence but with much less cultural identity and less crop diversity than before PROCAMPO. For the limited number of small farmers marketing their surplus production, payments from the program and the technical assistance provided have not been sufficient to increase the production of harvested products and achieve satisfactory integration into production and marketing chains [72]. The fact that PROCAMPO beneficiaries have used a significant portion of their payments to purchase chemical fertilizers, pesticides, and herbicides has not only degraded the quality (and taste) of the crops grown and consumed or sold by small farmer families but has also set in motion a vicious cycle in which program funds intended to improve the livelihoods of rural small farmers end up subsidizing industry and plunging small farmers deeper into economic and environmental crisis.

Despite these negative impacts, the PROCAMPO program has maintained its structure in the form of a conditional cash transfer program, which has been unchanged over a 28-year period. Sadly, PROCAMPO was not the only program of this scope that largely failed to achieve the goals it set for itself. Other programs, such as PRONADRI (1985–1988) (Figure 1), which preceded it, had the lofty goal of achieving food self-sufficiency through more active and organized participation of rural communities in planning and managing their own development. This included the allocation of transfers and subsidies to the agricultural sector, as mentioned in the Official Gazette of the Federation on 20 May 1985 [75]. Unfortunately, the balance of this program also left much to be desired. PRONADRI's "medium-term objectives, strategies and goals in the areas of social welfare, comprehensive agrarian reform, agricultural production, employment, and rural income, as well as sectoral policies for water, and forests," remained mostly unfulfilled [132–134]. The failure of both programs in fulfilling their promises points to the need for continuous monitoring of the program implementation process. That monitoring should in turn lead to incorporate effective strategies to address local needs and to predict, prevent, and assess their potential adverse effects.

In 2019, another ambitious CCT program, Sembrando Vida (Sowing Life), was launched, promising no less than lifting small farmers out of poverty and promoting pesticide-free agroecological production while reforesting 1 million hectares. Sembrano Vida pays small farmers MXN 6000.00 (USD 312; USD 1 = MXN 19.25) per month to plant trees and crops on 2.5 hectares. Unlike PROCAMPO, Sembrando Vida aims to enhance technical assistance through the establishment of CACS (peasant learning communities). These communities bring together small farmers, along with a social technician and a production technician, with the goal of promoting more effective guidance and training [77]. However, in just three

years, the program's approach to restoring forest cover through idle or abandoned areas has undergone significant changes in its operating rules [82,135–137]. This situation has generated confusion and controversy among beneficiaries nationwide. These disagreements revolve around the tree species that can be planted, the interpretation of concepts related to the type of forests that can be used as secondary growth, and the definition of abandoned lands [138–140]. This has even led to an increase in the incidence of wildfires in the Yucatán Peninsula [141]. In 2021, the planned budget for PROCAMPO, now called Production for Wellbeing, was MXN 13.5 billion (USD 6.7 million; USD 1 = MXN 20.29 in 2021) [142], whereas it was MXN 28.9 billion for Sembrando Vida (USD 14.3 million). However, the number of beneficiaries in PROCAMPO was 2,186,836 [143], whereas Sembrando Vida had only 449,936 beneficiaries [144]. This highlights the budgetary importance of Sembrando Vida over PROCAMPO as a new policy aimed at the Mexican agricultural sector and its natural resources.

Over the course of 28 years, PROCAMPO has made continuous changes in the implementation and operation of the program, and farmers have adapted to the conditions at hand. Now, within a very short period of time, Sembrando Vida, another CCT program, has been declared a nationwide program, and the conditions attached to the payment of these subsidies to small farmers have changed again. However, with respect to peasant farming practices and environmental protection, the neoliberal approach continues to prevail. Despite numerous promises from the government of a fundamentally new approach to Sembrando Vida, an evaluation conducted by CONEVAL in 2020 [145] indicated that the program lacks a diagnosis explaining the criteria for determining the coverage strategy and lacks a central objective that would go beyond simply measuring results in the delivery of aid. These factors could potentially lead to the failure of the program.

## 5. Conclusions

One lesson from this study is the need for a careful review of agricultural conditional cash transfer programs as a strategy for poverty alleviation, especially in view of their widespread use in Latin America. In Mexico, PROCAMPO has not addressed all of the steps involved in the agricultural production cycle: sowing, maintenance (e.g., weeding), harvest, commercialization, technology acquisition, training, etc. In Mexico's neoliberal context, most of the institutional framework to support agricultural production, such as the Agricultural Cooperative Bank (1930s) or CONASUPO (1960–1990s), which guaranteed fair prices, has disappeared [54,58,62]. Since the 1980s, government mechanisms to help small farmers improve their production systems through access to training and credit have also disappeared [1,2]. The conditions under which payments are granted focus solely on the amount of land under cultivation and its intensive use, encouraging the use of agrochemicals and pesticides to maintain production, thus jeopardizing the sustainability of contextually developed agricultural systems [74,96].

It would be desirable for programs focused on production and conservation in the Mexican countryside, such as PROCAMPO, to provide beneficiaries with access to fairer regional markets. Furthermore, such programs should ensure and actively support the maintenance of ecologically sound farming practices. Traditional farming practices are the result of longstanding traditions and local knowledge and are therefore optimally adapted to the agroecosystem in question. Because they involve maintaining forest as part of the system, traditional practices also participate in biodiversity conservation and the provision of many ecosystem services, Finally, when designed, such programs should consider transversality (social, economic, cultural, political, and biophysical identity) and assume implicit responsibility for the sustainable use of natural resources and their conservation.

**Author Contributions:** Conceptualization: L.-P.R., B.S., M.M.y.T.G.C. and S.C.; methodology: L.-P.R.; validation: B.S., M.M.y.T.G.C. and S.C.; formal analysis: L.-P.R.; data curation: L.-P.R. and C.M.-M.; writing—original draft preparation: L.-P.R. and B.S.; supervision: B.S. All authors participated in writing—review and editing. All authors have read and agreed to the published version of the manuscript.

**Funding:** This research was funded by the scholarship 626644 of Consejo Nacional de Ciencia y Tecnologia (CONACYT) granted to L.-P.R.

**Data Availability Statement:** No new data were created or analyzed in this study. Data sharing is not applicable to this article. The data presented in this study was obtained from the meta-analysis of papers already published see references [6,46,63,69,72–74,95–97,99,102–106,114–118,120].

**Acknowledgments:** We extend our gratitude to M.C. Holger Weissenberger for elaborating Figure 1 and to Susannah McCandless for the English editing of the entire manuscript.

**Conflicts of Interest:** The authors declare no conflict of interest. The funders had no role in the design of the study; in the collection, analyses, or interpretation of data; in the writing of the manuscript; or in the decision to publish the results.

## Appendix A

**Table A1.** Development of information about PROCAMPO over time (1994–2022).

| Variables | Categories | Indicators | Turner et al., 2001 (1969–1997) | Klepeis & Vance, 2003 (1986–1997) | Sadoulet, De Janvry & Davis, 2001 (1994–1997) | Abizaid & Coomes, 2004 (1998) | Dalle et al., 2006 (1980–2000) | Schmook, & Vance, 2009 (1997–2003) | Radel, Schmook & Chowdhury, 2010 (1997–2003) | Schwentesius et al., 2007 (2001–2003) | Bautista-Zúñiga & Mizrahi, 2005 (1997–2005) | Ellis, EA & Porter-Bolland, 2008 (2000–2005) | Winters & Davis, 2009 (2008) | SAGARPA & FAO, 2012 (1994–2009) | Arapau & Asociates, 2011 (1995–2009) | Krylov et al., 2019 (2000–2012) | Ellis, Romero & Hernández, 2015 (2001–2013) | CONEVAL, 2013 (2012–2013) | FAO & SAGARPA, 2015 (1993–2015) | Ellis, Gomez & Romero-Montero, 2017 (1980–2016) | Mardero et al., 2018 (1980–2016) | ASF, 2016 (1994–2016) | SAGARPA, 2017 (2014–2017) | Dobler-Morales, Chowdhury & Schmook, 2020 (2017–2018) |
|---|---|---|---|---|---|---|---|---|---|---|---|---|---|---|---|---|---|---|---|---|---|---|---|---|
| Farming practices | Economic dependence on subsidies | Dependence | | X | X | | | X | X | | | | | X | X | | | X | X | | X | X | X | |
| | Land use changes | Forest–agricultural change | | X | | X | X | X | X | | | | | X | | | X | | | | X | | | |
| | | Change from sisal and *milpa* areas to chili and pasture | X | X | | | | X | X | | X | | | X | X | | X | | | | X | | | |
| | Changes in agricultural techniques | Elimination of fallow | | X | | X | X | X | X | | X | X | | X | X | | | | | | X | X | X | X |
| | Production | Maize monoculture/lower diversity | | | | | X | | X | | | X | | X | | | X | | | X | | X | x | X |
| Indirect effects on conservation | Environmental degradation | Use of agrochemicals | | | | | | | | X | X | | | X | X | | | | | | X | X | | X |
| | | Water pollution | | | | | | | | X | | | | | | | | | | | X | | | |
| | Deforestation | Deforestation | X | X | | | X | X | X | | | | | X | X | X | X | | | X | X | | | |

X indicates the presence of indicators by category and variable in the documents reviewed.

## Notes

1    Special Program for Food Security (SPFS), created by FAO in Mexico in 1994 and aimed at supporting low-income food-deficit countries [10].

2    Strategic Support Project for the Corn and Bean Producers' Production Chain (PROMAF), which provides support for planting plots and purchasing improved seeds and fertilizers, and offers new forms of production and technical support [11].

3    The National Crusade Against Hunger (CNcH) was a 2013–2018 government strategy with the objective of improving the living conditions of millions of Mexicans in extreme food poverty [12].

4    Commercial Forestry Plantations Program (PRODEPLAN), created in 1997 through the Forestry Law as a strategy to strengthen the forestry sector [13].

5    The Secretariat of the Environment and Natural Resources (SEMARNAT) was created in 1994 in response to the need to plan the management of natural resources and environmental policies based on an integrated logic. Its initial name was the Secretariat of the Environment, Natural Resources and Fisheries (SEMARNAP) [14].

6    Ministry of Finance and Public Credit (SHCP), so called since 1953, with the purpose of overseeing the proper development of the Mexican financial system. It was founded, however, at the time of national independence on 27 September 1821 [15].

7    First it was the Secretariat of Agriculture and Hydraulic Resources (SARH) (1976–1994); then the Secretariat of Agriculture, Livestock, and Rural Development (SAGAR) (1995–2000); and then the Secretariat of Agriculture, Rural Development and Fisheries (SAGARPA) (2000–2018). Most recently, it is the Secretariat of Agriculture and Rural Development (SADR) [16].

8    The *milpa* is a traditional agroecosystem based on the planting of different crops, such as squash and beans. In addition, the *milpa* is usually interspersed with other edible herbs that grow naturally during the rainy season [21].

9    *Milperos* are farmers who belong mainly to the Mayan people, recognized for their culture and language. They usually obtain their seeds from the previous harvest but also potentially either inside or outside their community [33]. They sow *milpa* under different slash-and-burn variants. In the State of Yucatán alone, 45,000 traditional *milpera* families have been registered [34].

10   Even as an additional policy, whether a monetary or agricultural policy, guaranteed prices represent a state intervention in the marketing of the main agricultural food products by establishing a price to mitigate the irregularity of supply. This irregularity is due, among other reasons, to seasonal production and yields, which in turn depend largely on climatic conditions and the presence or absence of pests [60].

11   In relation to the supervision of PROCAMPO program resources, its decree establishes the following: "ARTICLE NINTH—The Secretariat, in collaboration with the Secretariat of the Comptroller General of the Federation, will implement the social comptroller scheme to involve producers in the supervision of the use of resources and actions carried out in PROCAMPO. To this end, the Steering Committees of the Rural Development Districts will promote the creation of Control and Surveillance Subcommittees in their respective geographic areas, as well as the election and training of social comptrollers from among the producers themselves" [71].

12   In 2001, for example, the average payment per cultivated hectare was MXN 1250.00 (USD 125; USD 1 = MXN 10) per year, according to Schwentesius and collaborators [97].

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
