# Peer review of "Public Policies Shaping Mexican Small Farmer Practices and Environmental Conservation: The Impacts of 28 Years of PROCAMPO (1994–2022) in the Yucatán Peninsula"

_land, doi:10.3390/land12122124_

Round 1
Reviewer 1 Report
Comments and Suggestions for Authors
This manuscript explores an interesting topic. However, there are some areas that need improvement.
1. Too many redundant references.
2. Lack of standardized empirical analysis. It is recommended to further analyze the spatiotemporal evolution relationship of the data.
3. It is needed to display the research area on a map.
4. Discussion section layout error. And it is necessary to focus on discussing the applicability of the paper outside of Mexico.
Comments on the Quality of English LanguageNeed polish.
Author Response
Please see the attachment with a detailed response to reviewer 1

Reviewer 2 Report
Comments and Suggestions for Authors
Minor editing of Novelty and significane of content required.
Comments on the Quality of English LanguageModerate editing of English language required.
Author Response
Please see the attachment with detailed responses to reviewer 2

Reviewer 3 Report
Comments and Suggestions for Authors
The introduction should be improved. please highlight the main issue of the study.
The literature review should be revised. The are a lot of articles that should be changed.
The methodology is ok.
For the result, the author should have justification and not just reporting.
The introduction should be improved. please highlight the main issue of the study.
The literature review should be revised. The are a lot of articles that should be changed.
The methodology is ok.
For the result, the author should have justification and not just reporting.
Author Response
Please see the attachment for a detailed response for reviewer 3

Reviewer 4 Report
Comments and Suggestions for Authors
This reviewer would like to thank the journal and the authors of the manuscript for the opportunity to review this interesting manuscript with title “Public Policies Shaping the Mexican Countryside: The Impact of 28 Years of PROCAMPO (1994-2022) on Small Farmer Practices and Environmental Conservation.”
This reviewer would like to convey the following three questions to the authors. These questions seek to support the authors to increase the likelihood of the manuscript, or an upcoming revised version of it, making further progress towards being eligible for publication in this journal.
Three questions:
1. Would the authors see value in mentioning in the title that the manuscript refers to the Yucatan Peninsula and not necessarily to the “Mexican Countryside” in general?
While the manuscript indicates in various sections (for example in the Abstract, Introduction, Materials and Methods) that the Yucatan Peninsula is the geographic focus of the manuscript, the title doesn’t mention the Yucatan Peninsula.
2. Would an additional section on “Conclusions” or “Concluding remarks” be of value?
Such section could have a style like the last one in current section 4. Discussion. Lines 797 to 805 in section:
“It would be desirable… (…)… conservation of agrobiodiversity.”
3. Would it be valuable for the manuscript to address in a somewhat more neutral way the positive and negative effects/ impacts of the PROCAMPO Program?, for example via listing/ documenting both the positive and negative effects, and then letting the likely predominance of the negative effects emerge after having explicitly address both the negative and positive effects?
At times, the manuscript nearly reads as if pre-assuming upfront that the PROCAMPO Program has had no positive effects, and sounds like concentrating on documenting just the negative effects.
For example, while section “4.2 Conservation effects” states in lines 584 – 586 “In its initial decree, PROCAMPO included environmental conservation and stated that it was necessary to contribute to the conservation of forests, reduce soil erosion and water pollution.”
soon after the manuscript seems to decisively concentrate on documenting the negative effects after the fragment in lines 591 – 601
“Despite these efforts to include this environmental conservation component discursively, the program had negative environmental impacts, including deforestation, during its implementation. For example, Ellis and Porter-Bolland (2008) showed that the reduction or elimination of fallow periods affected forest cover and forest recovery. (…)
Closing remark
This reviewer would like to thank again the co-authors of the manuscript and the journal for the opportunity to contribute to this review process, and wishes the co-authors all success in the upcoming steps of this review process.
Author Response
Please see attachment with a detailed response for reviewer 4

Reviewer 5 Report
Comments and Suggestions for Authors
Interesting manuscript that has potential - but at present reads more like a report or a dissertation. It can be much improved:
* mention in Abstract what is new in approach (method and/or theory)
* add more on the actual method used - yes literature from the networks like Red.... but how were things selected, tested/tabulated, etc.
* A map of Yucatan ?
* The discussion starts long before Discussion and there should perhaps be more sifting of info and analysis so things do not look like a report?
* There is no Conclusions - just a Discussion....
* Someone needs to check line-by-line and see that use of italic, mnetary units and program/Program are correct and consistent....
milpa often not italic Milperos ditto why $....pesos and later $ MXN
Use US$ or MXN and explain the latter at first use - some readers may be in India, etc. Milpa - explain this system at first mention by short note (is it just a shifting agric system)? What is cosmovision seems familiar but not a term I know.... cenote should it be italic?...92... not all spp names have been italicised Date of Mexican Revolution (readers may be in India, etc).
Author Response
Please see attachment with a detailed response for reviewer 5

Round 2
Reviewer 1 Report
Comments and Suggestions for Authors I have no further comments.Reviewer 5 Report
Comments and Suggestions for Authors
There is still a need the check Fig. 1 has Km SI units should be km
There are quite a few places where local terms are not italic e.g milpa in footnotes
Note spp. is not italicised normal print
Otherwise improved and I think now OK to proceed